# The Use of Drone Photo Material to Classify the Purity of Photovoltaic Panels Based on Statistical Classifiers

**DOI:** 10.3390/s22020483

**Published:** 2022-01-09

**Authors:** Tomasz Czarnecki, Kacper Bloch

**Affiliations:** The Institute of Telecommunications, Faculty of Electronics and Information Technology, Warsaw University of Technology, Nowowiejska 15/19, 00-665 Warsaw, Poland; 01112924@pw.edu.pl

**Keywords:** photovoltaic panel, detection, soiling

## Abstract

The subject of this work is the analysis of methods of detecting soiling of photovoltaic panels. Environmental and weather conditions affect the efficiency of renewable energy sources. Accumulation of soil, dust, and dirt on the surface of the solar panels reduces the power generated by the panels. This paper presents several variants of the algorithm that uses various statistical classifiers to classify photovoltaic panels in terms of soiling. The base material was high-resolution photos and videos of solar panels and sets dedicated to solar farms. The classifiers were tested and analyzed in their effectiveness in detecting soiling. Based on the study results, a group of optimal classifiers was defined, and the classifier selected that gives the best results for a given problem. The results obtained in this study proved experimentally that the proposed solution provides a high rate of correct detections. The proposed innovative method is cheap and straightforward to implement, and allows use in most photovoltaic installations.

## 1. Introduction

Since the 1950s, the temperature on Earth has risen on average by 0.2 C per decade. The temperature rise has a negative impact on the environment [1,2]. The consequences of climate warming include heatwaves deadly for humans, drinking water shortages, food production decline, coral reef degradation, and glacier melting [3]. To limit global warming, the European Union (EU) under the European Green Deal [4] initiated a strategy to achieve climate neutrality by 2050. The EU strategy has been endorsed by the European Parliament and the European Council, 2020 [5,6]. In order to achieve this goal, it is necessary to increase the use of renewable energy sources, which can achieve by building new renewable energy installations and increasing their energy efficiency.

Photovoltaics (PV) convert light into electricity using semiconductor materials that exhibit a photovoltaic effect. Solar cells are used to convert solar energy into electricity. The energy generated by solar cells is called “green energy”. This means that it comes from a natural and renewable energy source, the sun, and its production does not release pollutants into the air [7]. This assumption is valid, but only after approximately three years of operation of the PV module because the energy expenditure needed to produce it is returned after this time [8]. In addition, the production of energy used to make the panel generates 300 kg CO${}_{2}$ in the atmosphere. Advantages of photovoltaics compared to other types of RES are:–Positive correlation between the intensity of sunlight and the daily demand for electricity,–Increased generation in the summer period correlated with the demand for cold, and–It enables the use of brownfield sites and poor-quality land, as well as building roofs [9].

The TÜV Rheinland Institute identified the most common problems based on its data from photovoltaic farms (Figure 1), industrial installations, and home micro-installations. We include among them:–Dirt on PV panels,–Incorrect installation,–Shading,–Discoloration of EVA foil,–Glass breakage,–Degradation by induced voltage,–Path snails,–Defective protective foil,–Spot heating of panels [10].

The most common problem is dirty panels, which translates into huge losses in energy generated [11,12], as shown in 1. The dust accumulated on the surface of the photovoltaic panel comes mainly from the soil, rocks, construction debris, particles from car traffic, bird droppings, and pollen [13]. Dust accumulation on the surface of the panels obstructs the light, preventing it from reaching the PV cells, reducing energy production [14,15]. The energy loss depends on the amount of dust, particle size, and chemical composition of the powder. Contaminants have different effects on the light transmission process. Some dust particles can reduce the efficiency of photovoltaic devices by up to 98 [16,17]. To effectively produce electricity using PV cells, it is necessary to ensure failure-free operation of the PV installation throughout its lifetime (even up to 30 years), and a quick return on investment outlays. For this purpose, it is necessary to develop a fast, reliable, and straightforward method of checking the cleanliness of PV cells [18,19].

## 2. Requirements

The system (Figure 2), which the algorithm is to be a part of, consists of a drone equipped with a high-resolution camera, an edge computing unit, automatic cleaning devices, and, optionally, may also consist of cloud computing. The principle of the system is as follows. The drone uses a camera to record video material showing PV panels in a given area. The recorded video material is sent to a computing unit, which extracts individual frames from the video. It then analyzes them using the algorithm proposed, in this work, to detect dirty PV panels [20]. After detecting such a module, information about the need to clean the PV modules is sent to the cleaning devices. The photos collected can be sent to cloud computing for further analysis, e.g., to improve the algorithm’s operation.

The classification of a PV panel is to assign it to one of two classes (Figure 3):–Clean,–Dirty.

A clean panel works with the most excellent possible efficiency under the given atmospheric conditions, or the decrease in efficiency due to impurities on its surface is less than 25%. In turn, a dirty panel is one for which we notice a reduction in performance by at least 25%.

### 2.1. Drone Photo Sourcing

In the case of video materials obtained from a drone (digital sensors), spatial resolution, the Ground Sampling Distance (GSD) [21], is of crucial importance. It measures the distance in the field between the centers of two adjacent pixels (Figure 4). For each measurement mission of the drone, the GSD is defined before the task [22]. Typical values for this type of task should be in the range of 1.5 cm–45 cm. In the case of monitoring photovoltaic panels, it is necessary to determine which defects and the amount of dirt must be detected [23]. To detect panels with mechanical problems, set the GSD within 25 cm. To detect physical damage or points more minor than the entire panel, set the GSD between 5–16 cm. In the case of dust and dirt detection missions, set the GSD > 2 cm—this will allow you to detect even small contaminants on the panels [24]. The tested system uses RGB cameras with a resolution of 20 MP and a 1/1.7″ CMOS matrix (DJI Zenmuse H20).

### 2.2. The Research Material

The input data will be the photographic material recorded on the test stand reflecting the actual conditions on the photovoltaic farm. The photographic material consists of 70 photos encoded in JPG format. The images show one, two, three, or four solar panels. The detailed breakdown is as follows:–60 photos containing one panel,–4 photos containing two panels,–4 photos containing three panels,–2 photos contain four panels.

The research material contained a total of four different polycrystalline PV panels, 44 clean and 44 dirty panels in total. First, pictures of clean PV panels were taken, then the dirt from the soil, rocks, construction debris, and particles from car traffic was gradually applied to them, and at a power drop of 10%, a photo was taken until archived a power drop of 50% compared to maximum power.

The images were taken:–With adequate sunlight. The images were taken during the day with a minimum solar radiation intensity of 500 W/$m^{2}$, because below this value the PV panels are insufficiently illuminated, which means that the contrast of the photo is too low to extract the information that is important to us. The project does not assume artificial lighting of PV panels.–Under appropriate weather conditions. Pictures cannot be taken during rainfall, as they introduce unwanted artifacts into the picture, making subsequent analysis difficult.–At a minimum angle of 45${}^{{}^{\circ}}$ to the panel surface. Smaller values may make it impossible to extract the panel from the photo.–At different times of the year. This approach will enable the use of classifiers throughout the year.

## 3. Statistical Classifiers to Classify Photovoltaic Panels

The project presents several algorithm variants that use different statistical classifiers to classify photovoltaic panels into one of two classes, clean or dirty, based on observation (feature vector). For the classification, it decided to use the characteristic feature of dirt. The image saturation decreases in the place where they occur, the image saturation decreases, and its luminance increases. Due to the two classes and two features, binary classifiers have been selected to classify observations with two parts. The algorithm consists of two stages: classifier training and classification. The photo material was divided into two sets: a training set composed of 32 photos of clean panels and 32 photos of dirty panels. A test set consisted of 12 pictures of clean panels and 12 photos of dirty panels. Before classification, the image is pre-processed to remove unwanted background and leave only the PV module in the picture.

### 3.1. Extracting a Panel from a Photo

The first step in classification is to detect the PV module in the image. The PV surface is found in the image and extracted from the background at this stage. Any additional information is dropped from the input image.

This process takes place in three steps:–Detection of all edges in the photo,–Finding the edge of the PV panel,–Application of a forward-looking transformation [25].

The result of each of these steps is shown in Figure 5.

### 3.2. Observation

By visually comparing a clean and dirty panel, you can see that the surface of the dirty panel looks dull and lacks color intensity. This is because the dominant color of the dust is gray, and the shades of gray are not saturated. In addition, the surface of the pure solar panel is dark as the cell material absorbs incident light. Therefore, the amount of light reflected by the panel is limited. For a dirty PV module, its surface looks brighter because less light is absorbed by the cell, and therefore more light is reflected from the surface and scattered by the deposits [26]. Table 1 shows the average values of color saturation and luminance for the same panel in two cases, when clean and dirty. Figure 6 shows a visual comparison of this panel. These panels differ mainly in appearance. No structure is visible, no light reflection, surface heterogeneity. In turn, Figure 7 presents all the results of observations in a graphical form. These values confirm the hypothesis that for dirty panels, the image saturation decreases, and the luminance increases.

Image luminance is a value representing the image’s brightness, calculated by ITU BT.601 [27]. For each pixel, calculate its luminance value and then calculate the arithmetic mean of these values.

To calculate the saturation value of the image, you can use the fact that the saturation is one of the components of the HSV model. So you can convert the image from RGB to HSV, extract the component corresponding to the saturation for each pixel, and then calculate the arithmetic mean of these values.

### 3.3. The Classifier of the k Nearest Neighbors

The classifier of the *k* nearest neighbors (kNN) classification method assigns a classified object to the class that is most frequently represented among the *k* closest neighbors from the training set [28]. In order to find the *k* nearest neighbors of the test object, the Euclidean distance between the test object and all training objects is calculated [29]. In this case, the classifier consists of two features, so it is two-dimensional, which means that can lace of the things be placed on the Euclidean plane in the form of a point with Cartesian coordinates. Then, the Euclidean distance between two points is expressed by the formula:(1)$$d\left(p,q\right)=\sqrt{{\left(p_{1}-q_{1}\right)}^{2}+{\left(p_{2}-q_{2}\right)}^{2}}$$
where:

*p*—first point;

*q*—second point;

d(p,q)—Euclidean distance between points *p* and *q*;

$p_{1}$—coordinate *X* of point *p*;

$p_{2}$—coordinate *Y* of point *p*;

$q_{1}$—coordinate *X* of point *q*;

$q_{2}$—coordinate *Y* of point *q*.

As shown in Table 1, the values of saturation and contrast differ significantly in terms of magnitude, so the difference would dominate the calculated Euclidean distance because it affects the distance value much more. For this reason, it is necessary to normalize the value so that all dimensions for which the distance is calculated are equally relevant. Normalization consists of making the variable’s values belong to the interval [0, 1]. The formula that expresses it:(2)$$x_{j}\left(i\right)=\frac{x_{j}\left(i\right)-min\left(x_{j}\right)}{max\left(x_{j}\right)-min\left(x_{j}\right)}$$
where:

*i*—next vector index;

*j*—index of feature;

$max(x_{j})$—the maximum value of the variable *j*;

$min(x_{j})$—the minimal value of the variable *j*.

Test data also needs to be normalized. When normalizing the test set, one should use the maximum and minimum values determined on the training set. Figure 8 shows the decision surface for the kNN classifier for *k* = 7.

### 3.4. Naive Bayesian Classifier

The naive Bayes classifier is a simple probabilistic classifier that assumes that all features are mutually independent, hence the so-called “naivety” of this classifier. It uses Bayes’ theorem, and the classification result is based on a conditional probability comparison. The class for which the posterior probability value is the highest is selected [30].
(3)$$P\left(Y_{k}|X\right)=\frac{P\left(X|Y_{k}\right)P\left(Y_{k}\right)}{P(X)}$$
where:

*Y*—vector of classes;

$Y_{k}$—class;

*X*—vector of features of classified object;

$X_{i}$—feature;

$P(Y_{k})$—probability *a priori*;

P(X|Yk)—probability of occurrence;

$P(Y_{k}|X)$—probability *a posteriori*;

P(X)—probability of occurrence of set of features.

Using the assumption of the classifier’s naivety:(4)$$P\left(X|Y\right)=\prod_{i=1}^{n}P\left(X_{i}|Y\right)$$
where:

$P(X|Y_{k})$—probability of occurrence;

$P(X_{i}|Y)$—conditional probability of occurrence of a given feature provided that a given class occurs.

The values of the features are continuous, so we assume that for each part $X_{i}$ the distribution $P(X_{i}|Y_{k})$ is a normal distribution:(5)$$P\left(X_{i}|Y_{k}\right)=N\left(\mu_{ik},\sigma_{ik}^{2}\right)=\frac{1}{\sqrt{2\pi \sigma_{ik}^{2}}}e^{-\frac{1}{2\sigma_{ik}^{2}}{\left(X_{i}-\mu_{ik}\right)}^{2}}$$
(6)$$\mu_{ik}=\frac{1}{\sum_{j=1}^{m}1_{Y}\left(Y^{j}=Y_{k}\right)}\sum_{j=1}^{m}X_{i}^{j}1_{Y}\left(Y^{j}=Y_{k}\right)$$
(7)$$\sigma_{ik}^{2}=\frac{1}{\sum_{j=1}^{m}1_{Y}\left(Y^{j}=Y_{k}\right)}\sum_{j=1}^{m}\left(X_{i}^{j}-\mu_{ik}^{2}\right)1_{Y}\left(Y^{j}=Y_{k}\right)$$
where:

$1_{Y}$—indicator function;

$\sigma_{ik}^{2}$—variance;

$\mu_{ik}$—average value.

In the case under consideration, we have 2 features and 2 classes:(8)$$X=[X_{saturation},X_{luminance}]$$
(9)$$Y=[Y_{clean},Y_{dirty}]$$
therefore:(10)$$P\left(Y_{clean}|X\right)=\frac{P\left(Y_{clean}\right)*P\left(X_{saturation}|Y_{clean}\right)*P\left(X_{luminance}|Y_{clean}\right)}{P(X)}$$
(11)$$P\left(Y_{dirty}|X\right)=\frac{P\left(Y_{dirty}\right)*P\left(X_{saturation}|Y_{dirty}\right)*P\left(X_{luminance}|Y_{dirty}\right)}{P(X)}$$

To compare $P(Y_{\mathrm{clean}}|X)$ and $P(Y_{\mathrm{dirty}}|X)$, it is not necessary to calculate *P*(*X*) because this value is constant and only serves as a scaling function. This approach reduces the computational effort of the classifier. The classifier’s decision rule is as follows:(12)$$Y_{k}=\left\{\begin{array}{cc} clean & \mathrm{if}P\left(Y_{\mathrm{clean}}|X\right)⩾P\left(Y_{\mathrm{dirty}}|X\right)\\ dirty & \mathrm{otherwise} \end{array}\right.$$

Figure 9 shows the decision surface of the naive Bayes classifier.

### 3.5. Fisher’s Linear Discriminator

The Fisher Linear Discriminator (FLD) is used for supervised classification and produces a linear discriminant rule. The task of discriminant analysis for two classes can be defined as [31]:

find the direction *a*, hat best separates the learning subgroups, and as a measure of class separation along a given direction *a* take the square of the distance between the arithmetic means of the subgroups along this direction, taking into account the variability of the intra-group observation.


(13)
$$\begin{array}{c} \frac{{\left(a^{\prime }{\bar{x}}_{2}-a^{\prime }{\bar{x}}_{1}\right)}^{2}}{a^{\prime }Wa} \end{array}$$


The direction of *a* best separating the classes is the direction that maximizes the expression (Equation 13):(14)$$\begin{array}{c} argmax_{a}\frac{{\left(a^{\prime }{\bar{x}}_{2}-a^{\prime }{\bar{x}}_{1}\right)}^{2}}{a^{\prime }Wa} \end{array}$$

The solution is:(15)$$a=W^{-1}\left({\bar{x}}_{2}-{\bar{x}}_{1}\right)$$
where:

*W*—intragroup covariance matrix;

*a*—direction vector of the searched line;

${\bar{x}}_{1}$—group mean of observations included in the class *clean*;

${\bar{x}}_{2}$—group mean of observations included in the class *dirty*;

$n_{1}$—number of observations included in the class *clean*;

$n_{2}$—number of observations included in the class *dirty*.

The observations can be divided into a subgroup of observations classified as class *clean* and into a subgroup of observations classified as class *dirty*:$$\begin{array}{c} x_{11},x_{12},\cdots ,x_{1n_{1}}\mathrm{observations}\mathrm{from}\mathrm{class}\mathrm{clean}\\ x_{21},x_{22},\cdots ,x_{2n_{2}}\mathrm{observations}\mathrm{from}\mathrm{class}\mathrm{dirty} \end{array}$$

Then, we can write the group averages as:(16)$${\bar{x}}_{k}=\frac{1}{n_{k}}\sum_{i=1}^{n_{k}}x_{ki}\mathrm{dla}k=1,2.$$

In order to assess the intragroup variability of the covariance matrix, it is necessary to assume that both subgroups have the same covariance matrix, then:(17)$$W=\frac{1}{n-2}\sum_{k=1}^{2}\left(n_{k}-1\right)S_{k}=\frac{1}{n-2}\sum_{k=1}^{2}\sum_{l=1}^{n_{k}}\left(x_{kl}-{\bar{x}}_{k}\right){\left(x_{kl}-{\bar{x}}_{k}\right)}^{\prime }$$
where:



$n=n_{1}+n_{2}$



$S_{k}$—sample covariance matrix of subgroup *k*;

$x^{\prime }$—vector transposition *x*.

Having the designated direction *a*, both means of classes ${\bar{x}}_{1}$
*i*
${\bar{x}}_{2}$ and the new observation *x*, we can define the classification rule:(18)$$X=\left\{\begin{array}{cc} clean & \mathrm{if}\;|a^{\prime }x-a^{\prime }{\bar{x}}_{1}\left|<\right|a^{\prime }x-a^{\prime }{\bar{x}}_{2}|\\ dirty & \mathrm{otherwise} \end{array}\right.$$
where:

*x*—new observation;

*X*—the class assigned to the new observation.

What after qualifying the boundary case to class *clean* comes down to the following decision rule:(19)$$X=\left\{\begin{array}{cc} clean & \mathrm{if}{\left({\bar{x}}_{2}-{\bar{x}}_{1}\right)}^{\prime }W^{-1}\left[x-\frac{1}{2}\left({\bar{x}}_{1}+{\bar{x}}_{2}\right)\right]⩾0\\ dirty & \mathrm{otherwise} \end{array}\right.$$

Figure 10 shows the discriminant line. It is a straight line perpendicular to the line *a* and passing through the middle of the line connecting points $a{\bar{x}}_{1}$ and $a{\bar{x}}_{2}$. The discriminant line equation is as follows:(20)$$x_{L}=170,5716401240x_{N}+44,975372529300$$
where:

$x_{N}$—value of saturation;

$x_{L}$—value of luminance.

Figure 11 shows the designated decision areas.

## 4. Analysis and Research

The aim of the work was to test the quality and effectiveness of the proposed classifiers. For this purpose, the classic metrics of binary classification were used [32]. The decision surfaces of classifiers were also analyzed. In the case of the kNN classifier, its various variants were analyzed to select the optimal value of the *k* parameter.

### 4.1. kNN Classifier

TThe kNN classifier is characterized by the fact that for a new observation, the class that is most frequently represented among the *k* closest neighbors from the training set is selected. The correct performance of this classifier depends on the number of neighbors. The problem is the difficulty of selecting a priori the appropriate value for the *k* parameter, so different values of this parameter were analyzed. Only odd values of *k* have been considered, as such values guarantee that there will be no draw situation.
(21)k∈{1,3,5,7,9,11,13}

Table 2 presents the results of the classification for different values of the *k* parameter. Based on these results, classic binary classification metrics have been developed, which can be found in the Table 3. The test set consisted of 12 clean panels and 12 dirty panels.

It can be concluded that the optimal value of the *k* parameter is 7. For this value, the kNN classifier is characterized by the highest sensitivity, specificity and precision. Additionally, the F1 metric reaches the highest value for *k* = 7, which can be seen in the Figure 12.

### 4.2. Decision Surfaces

Figure 13 shows the decision surfaces of the classification of the 7NN classifier, the naive Bayes classifier and the linear Fischer discriminant. In order to compare these surfaces, the luminance and saturation values have been normalized so that, for each classifier, the values are determined from the same selected range.

Comparing the decision surfaces of the naive Bayes classifier and the linear Fischer discriminant, it can be concluded that these surfaces are similar to each other. Although the naive Bayes classifier is not a linear classifier, for this classifier, the boundary between two classes is similar to a straight line and resembles the boundary of a linear Fischer discriminant. In the case of the kNN classifier, its decision boundary is more wavy than for the other classifiers.

### 4.3. Metrics and Results

To test the quality and effectiveness of the proposed classifiers, traditional binary classification metrics were used. The following classic binary classification metrics were used to assess the correctness of the type: TPR, TNR, PPV, NPV, and F1 [33]. Table 4 presents the results of panel classification into classes clean and dirty. The individual lines contain the results for each of the tested classifiers. The columns provide information on the number of clean and dirty panel samples tested, the number of true positives and true negatives detected, and the number of false positives and false negatives seen.

Based on the results from Table 4, the classic binary classification metrics have been developed, which are presented in Table 5. The lines of this table present the metrics of each classifier, and the individual columns contain information about TPR, TNR, PPV, NPV, and F1.

Based on the research, it can be concluded that the Naive Bayesian classifier is characterized by the highest efficiency of detection of contaminated panels. All metrics for this classifier are the highest and amounted to 92%. For the 7NN classifier, the TPR, PPV, NPV, and F1 values are lower than for the Naive Bayes classifier, and the TNR value is the same. As for the linear Fischer discriminator, the TNR, PPV, NPV, and F1 values are lower than for the Naive Bayes classifier, and the TPR value is the same. Figure 14 shows.

The conducted research shows that the naive Bayes classifier is the optimal classifier for a given problem. A very high sensitivity characterizes the Bayes classifier. This means that it identifies clean panels with high efficiency. It also identifies dirty panels with high efficiency, as the specificity of this classifier is also very high. This classifier is also very precise. As for the Fischer discriminator, it is characterized by high sensitivity but low specificity, which means that it identifies clean panels more effectively than dirty panels. It is also exact in detecting clean panels, but has low precision in detecting dirty panels. In the case of the 7NN classifier, the situation is the opposite. This classifier has a high specificity and low sensitivity, identifying clean panels more efficiently than dirty panels. It is characterized by low precision in detecting clean panels and high precision in the detection of dirty panels.

## 5. Summary

Monitoring the cleanliness of photovoltaic panels is very important. In the first three years, the drop in efficiency can be as much as 15%. In places with high industrialization or dusty environments, the reductions in inefficiency are even more significant. Scientific research conducted by H. Haberlin and C. Renken from Berne University of Applied Sciences show that regular cleaning of PV modules improves their efficiency by up to 13.8% [34]. This solution is in line with the global trend of optimizing the use of photovoltaic panels. The results obtained in this study showed experimentally that the proposed solution provides a high rate of correct detections. The proposed innovative method is cheap and straightforward to implement, which allows it to be used in most photovoltaic installations and is suitable for use in an intelligent system for monitoring the cleanliness of photovoltaic panels. The presented methods of classifying the cleanliness of photovoltaic panels work well in areas with the highest concentration of dust and pollution. These are mostly suburban areas, proximity to highways, industrial plants, areas with a strong dusting of plants. There is no restriction in use. You can use photo material from a photovoltaic farm drone and photos with an appropriate resolution of home panels, e.g., on the roof, taken with a camera or telephone.

## Figures and Tables

**Figure 1 sensors-22-00483-f001:**
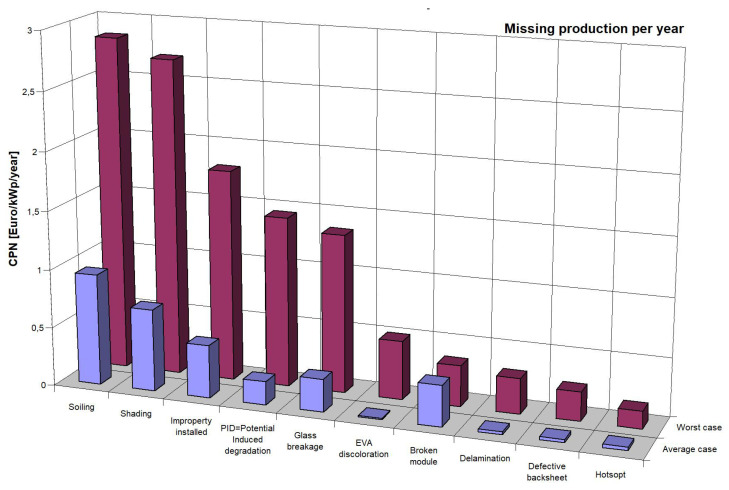
Missing electricity production due to various types of failure of solar panels [10].

**Figure 2 sensors-22-00483-f002:**
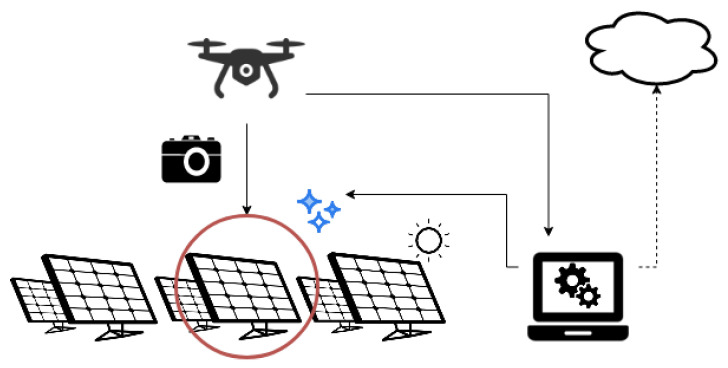
System.

**Figure 3 sensors-22-00483-f003:**
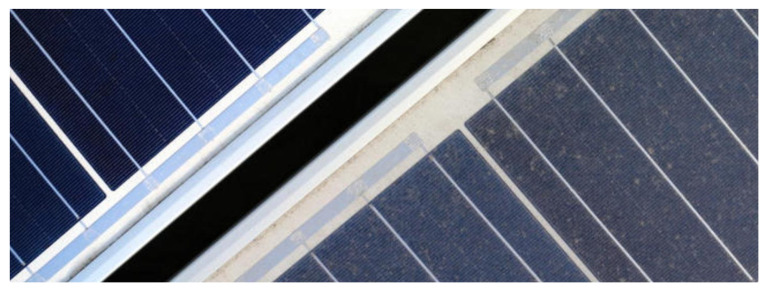
The classification of a PV panel—one of two classes.

**Figure 4 sensors-22-00483-f004:**
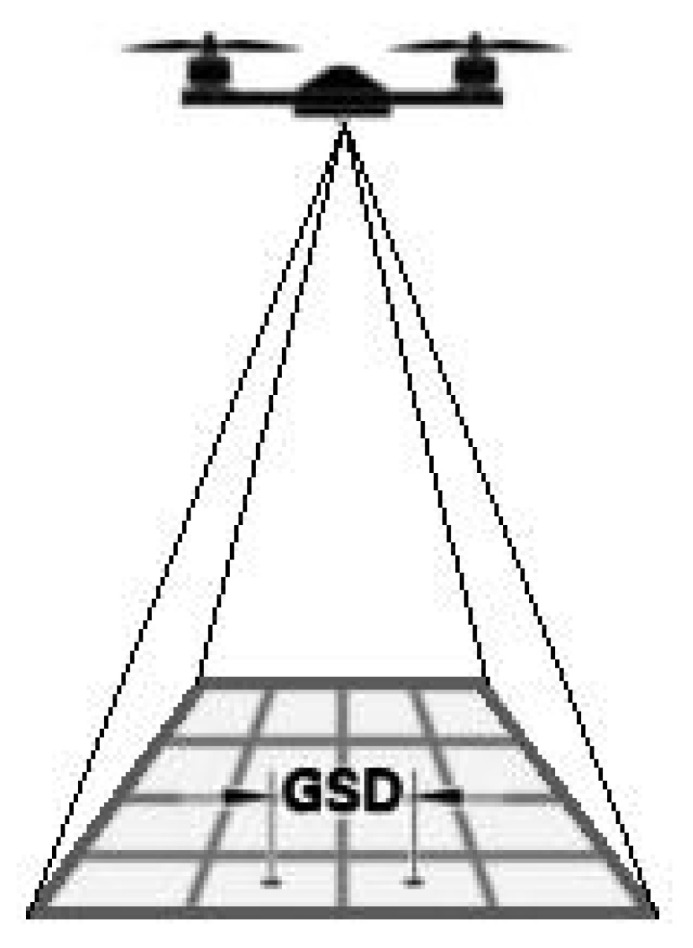
Ground sampling distance definition.

**Figure 5 sensors-22-00483-f005:**
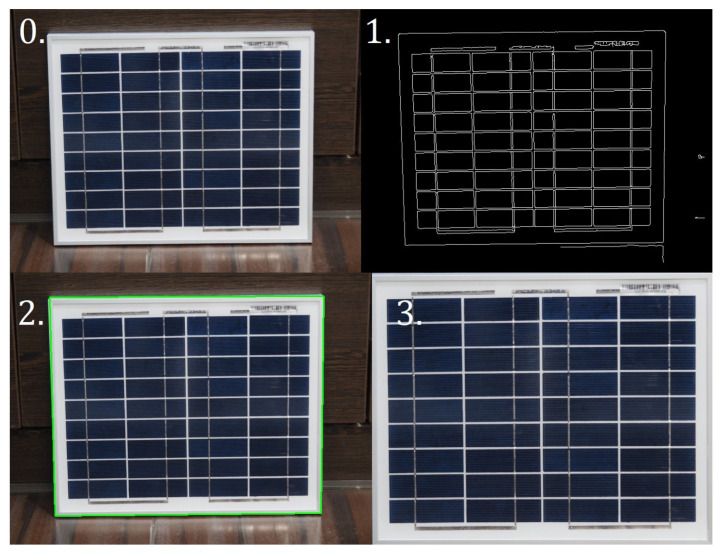
Successive stages of extracting the PV panel from the photo. 0—starting image; 1—detected edges in the picture; 2—outer edges of the PV panel found; 3—the result of the perspective transformation.

**Figure 6 sensors-22-00483-f006:**
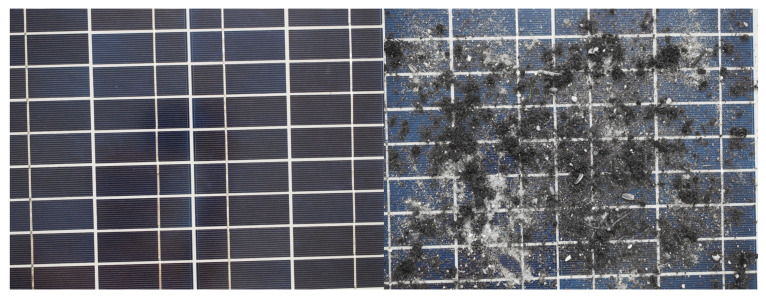
A clean panel (**left**) and a dirty panel (**right**) a comparison.

**Figure 7 sensors-22-00483-f007:**
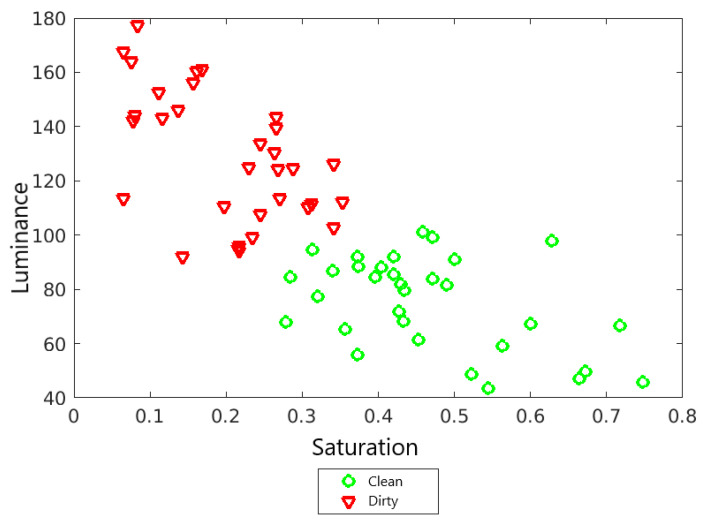
Observation results.

**Figure 8 sensors-22-00483-f008:**
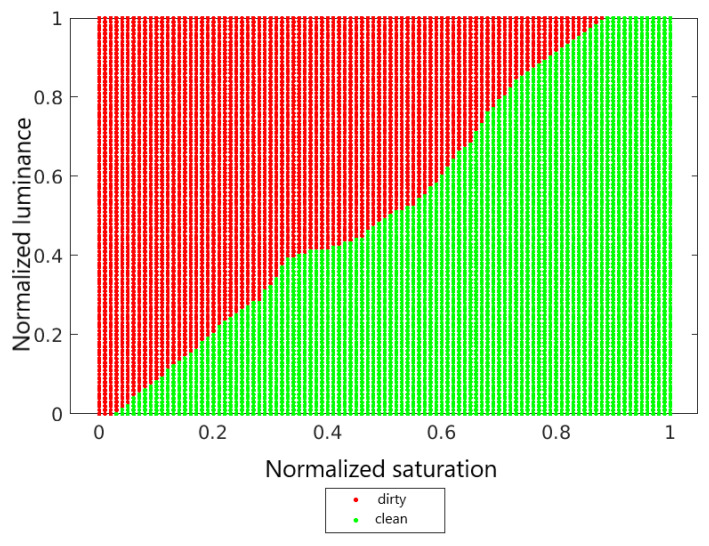
Decision area for the kNN classifier for *k* = 7.

**Figure 9 sensors-22-00483-f009:**
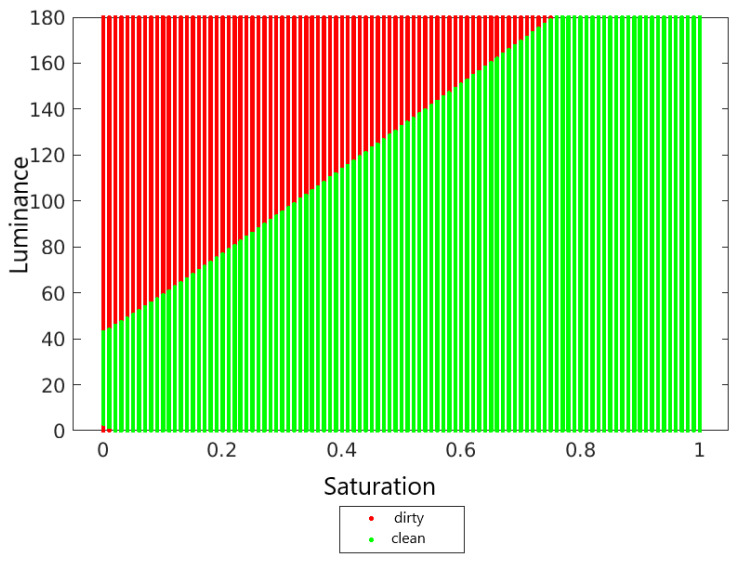
Decision surface for the naive Bayesian classifier.

**Figure 10 sensors-22-00483-f010:**
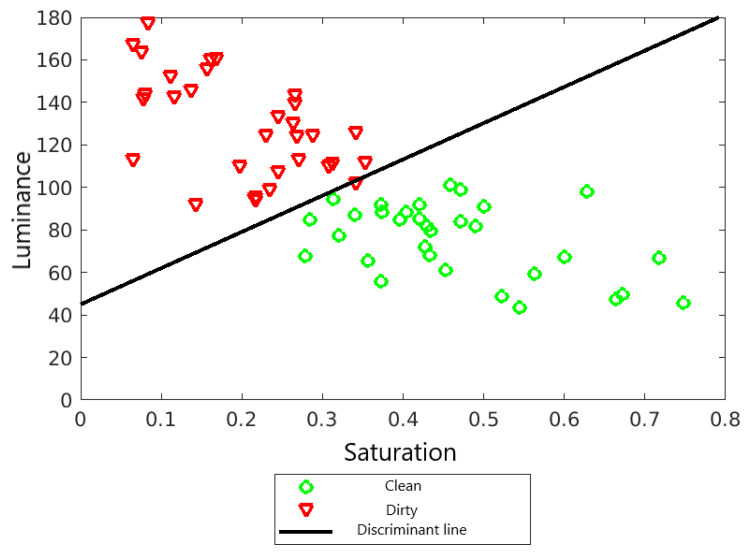
Decision boundary between classes.

**Figure 11 sensors-22-00483-f011:**
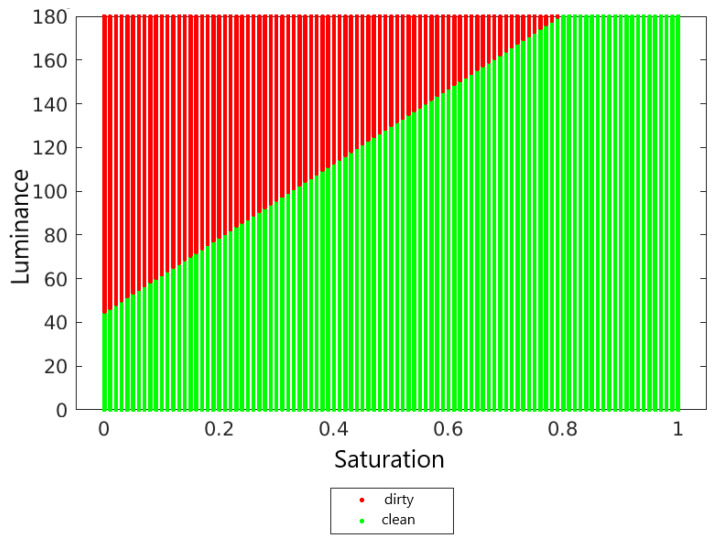
Decision surface for the linear Fischer discriminant.

**Figure 12 sensors-22-00483-f012:**
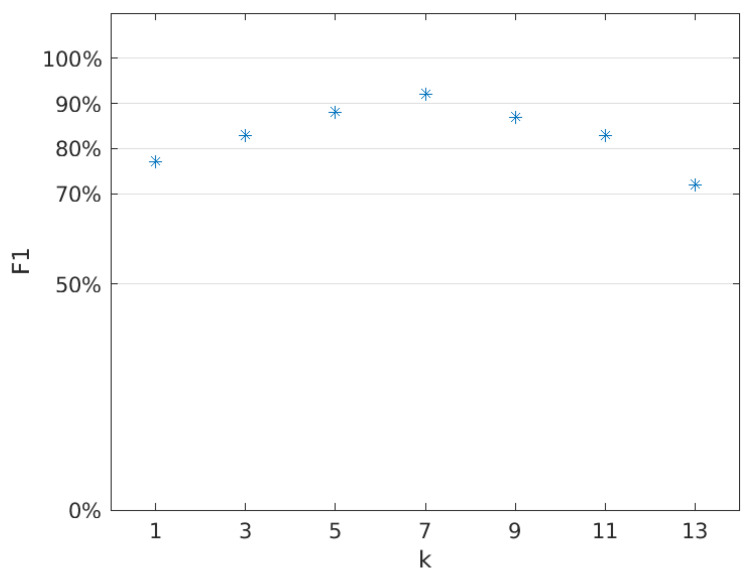
The value of the F1 metric depending on the parameter *k*.

**Figure 13 sensors-22-00483-f013:**
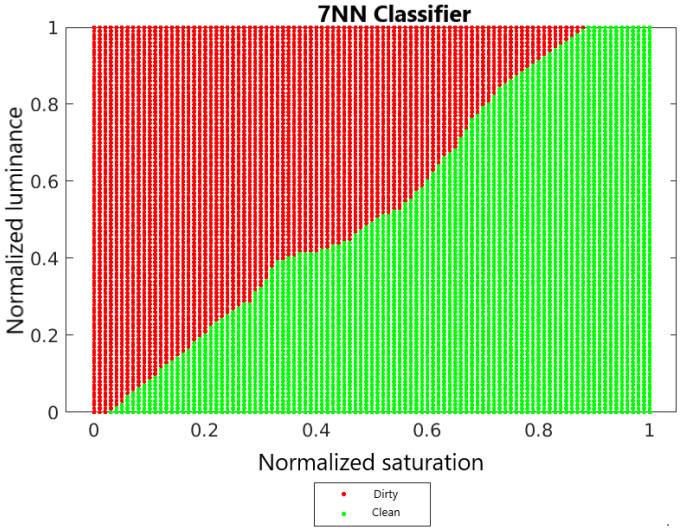
Decision surfaces of each classifier.

**Figure 14 sensors-22-00483-f014:**
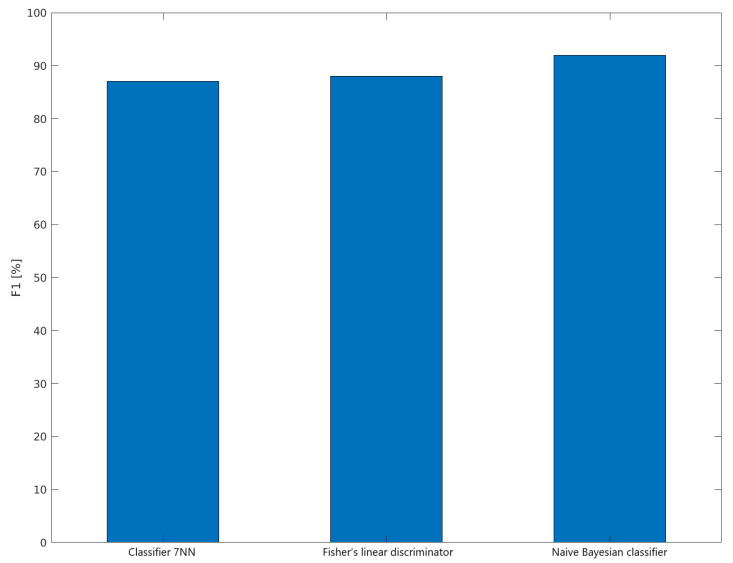
Value of the F1 metric for each of the classifiers.

**Table 1 sensors-22-00483-t001:** Average values of saturation and luminance for a clean and dirty panel.

	Clean	Dirty
Saturation	0.4707	0.0891
Luminance	84.3143	143.0543

**Table 2 sensors-22-00483-t002:** Classification results for different values of parameter *k*.

*k*	P	N	TP	TN	FP	FN
1	12	12	10	8	4	2
3	12	12	10	10	2	2
5	12	12	11	10	2	1
7	12	12	11	11	1	1
9	12	12	10	11	1	2
11	12	12	10	10	2	2
13	12	12	9	8	3	2

**Table 3 sensors-22-00483-t003:** Classic binary classification metrics for different values of parameter *k*.

*k*	TPR	TNR	PPV	NPV	F1
1	83%	67%	71%	80%	77%
3	83%	83%	83%	83%	83%
5	92%	83%	85%	91%	88%
7	92%	92%	92%	92%	92%
9	83%	92%	91%	85%	87%
11	83%	83%	83%	83%	83%
13	75%	67%	69%	73%	72%

**Table 4 sensors-22-00483-t004:** Classification results.

Classifier	P	N	TP	TN	FP	FN
The classifier of the *k* nearest neighbors	12	12	10	11	1	2
Naive Bayesian classifier	12	12	11	11	1	1
Fisher’s linear discriminator	12	12	11	10	2	1

**Table 5 sensors-22-00483-t005:** Classic metrics for binary classification.

Classifier	TPR	TNR	PPV	NPV	F1
Classifier 7NN	83%	92%	91%	85%	87%
Naive Bayesian classifier	92%	92%	92%	92%	92%
Fisher’s linear discriminator	92%	83%	85%	91%	88%

## Data Availability

Not applicable.

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
