# Peer review of "The Use of Drone Photo Material to Classify the Purity of Photovoltaic Panels Based on Statistical Classifiers"

_sensors, 2022, doi:10.3390/s22020483_

Round 1
Reviewer 1 Report
The authors propose a new method to detect dirt on photovoltaic panels. The problem is important and the authors' idea is clear. However, many essential details in the methods are missing (for example camera model and resolution, altitude of the drone). Moreover, I miss references to similar methods published before that dealt with this problem. The authors should present them in detail and then identify a gap. I think this paper may be accepted for publication if those problems are solved. In addition, I have more minor comments in the attached pdf file.

Author Response
The presented methods of classifying the cleanliness of photovoltaic panels work well in areas with the highest concentration of dust and pollution. These are primarily suburban areas, proximity to highways, industrial plants, areas with a substantial dusting of plants. As for the possibilities of their use, there are no restrictions. You can use photo material from a photovoltaic farm drone and photos with an appropriate resolution of home panels, e.g., on the roof, taken with a camera or telephone. The amount of energy saved depends very much on the quality of the solar panels. Comparative studies are underway. The approach to this topic is different. Most often, the decline in the efficiency of the entire photovoltaic farm, not the panel itself, is defined.
Moderate English changes required - corrected.
Title proposal: "The use of drone photo material to classify the purity of photovoltaic panels based on statistical classifiers"
Reviewer 2 Report
Here are some comments and recommendations:
- The reference to a figure should be “Figure 1”, not just “1”; same for Figure 3
- There is no reference to Fig.2
- It would be interesting to introduce the photos of a clean and dirty panel (Figure 4) at the beginning of chapter 3.2 and make clearer what the differences are.
- In Eq.2 and its explanations, you missed to close some parenthesis
- Please justify the selection of k=7 in your example, although the reason can be found at the end of the article.
- The names of the chapters could be improved, to make it clear where the case study begins.
- The English language should be checked, to make the text clearer and more readable; the abstract should be improved
- “lub” seems not an English word, please translate it from polish
Author Response
Moderate English changes required - corrected.
The abstract should be improved - done.
It would be interesting to introduce the photos of a clean and dirty panel (Figure 4) at the beginning of chapter 3.2 and make clearer what the differences are - explanatory notes added.
Please justify the selection of k=7 in your example, although the reason can be found at the end of the article - The parameter k was not predetermined. It is the result of research.
The names of the chapters could be improved, to make it clear where the case study begins - there are corrections.
Round 2
Reviewer 1 Report
The manuscript has improved and the remaining comments (attached) are minor. However, they should be implemented before the paper is accepted for publication.

Author Response
Recommended corrections made, literature added.